# Interdisciplinary Assessment of Premature Newborns and Their Families in a Hospital Setting in Medellín, Colombia

**DOI:** 10.3390/children12111483

**Published:** 2025-11-03

**Authors:** Juan Esteban López Cardona, Angie Estefanía Mesa Burbano, Leidy Yohana Apolinar Joven, Jenny Paola Ojeda Casallas, Natalia Pérez Doncel, Jhonatan Smith García Muñoz

**Affiliations:** 1Psique y Sociedad, Programa de Psicología, Fundación Universitaria María Cano, Medellín 050021, Colombia; 2Fisioter, Programa de Fisioterapia, Fundación Universitaria María Cano, Medellín 050021, Colombia; angieestefaniamesaburbano@fumc.edu.co (A.E.M.B.); leidyyohanaapolinarjoven@fumc.edu.co (L.Y.A.J.); 3Fonotec, Programa de Fonoaudiología, Fundación Universitaria María Cano, Medellín 050021, Colombia; jennypaolaojedacasallas@fumc.edu.co; 4Clínica Universitaria Bolivariana, Maternofetal, Medellín 050021, Colombia; natalia.perezd@upb.edu.co; 5Programa de Estadística, Universidad Nacional de Colombia, Medellín 050021, Colombia; jhsgarciamu@unal.edu.co

**Keywords:** premature, cerebral palsy, interdisciplinary studies, early diagnosis, quality of life, deglutition

## Abstract

Background: Preterm infants are highly fragile and at increased risk of developing Cerebral Palsy (CP). Therefore, early detection through an interdisciplinary approach is necessary to enable timely referrals and evidence-based interventions. The literature recommends the use of the Hammersmith Infant Neurological Examination (HINE), the WHOQOL-BREF quality of life questionnaire, and the Comprehensive Neonatal Speech-Language Assessment Protocol (EFIN) for early CP diagnosis. However, despite the availability of these tools, they have not yet been implemented as part of evaluation and follow-up protocols in Colombia. Methods: A cross-sectional observational and analytical study was conducted to analyze, in a group of preterm infants, the relationship between neurological risk, primary stomatognathic functions (suction-swallowing-breathing), and caregivers’ perceived quality of life. Results: A total of 43 preterm infants were included. Of these, only 9.30% showed neurological risk; 97.67% did not present alterations in the suction-swallowing-breathing triad; and the lowest quality of life scores were reported in social relationships and psychological health. Conclusions: There are perinatal factors that require follow-up in preterm infants to prevent possible future complications. It is essential to address both social and psychological aspects in family support programs.

## 1. Introduction

Each year, approximately 15 million children are born prematurely worldwide, representing 10% to 15% of births. In Colombia, the annual incidence is 8%, which accounts for about 50% of childhood disability cases [1].

Preterm newborns are highly vulnerable, as they face an increased risk of respiratory, cardiovascular, and neurological complications, which in some cases can lead to brain injury. Many require admission to Neonatal Intensive Care Units (NICU), a situation that affects their motor, emotional, social, and cognitive development, as well as the well-being of their families.

Specialized care for these infants also generates high costs for the health system, while the prediction of future neuromotor conditions remains challenging, particularly in relation to cerebral palsy (CP) [2].

Preterm birth is associated with a higher risk of CP [3]. Perinatal hypoxia is one of the main causal factors, responsible for about 23% of the 4 million neonatal deaths worldwide [4]; Nevertheless, it is not the only determinant. Rees et al. (2022) emphasize additional risk factors such as intraventricular hemorrhage, congenital infections, brain malformations, intrapartum complications, and genetic alterations [5].

These conditions have a long-term impact on the child’s functionality, leading to varying degrees of disability that limit independence and quality of life. The most affected areas are self-care, mobility, communication, and social interaction [5].

Early detection of CP is essential, as it enables timely referrals and the implementation of evidence-based interventions. Such early care maximizes neuroplasticity, improves functionality, and enhances family well-being [6]. To achieve this, valid and reliable assessment tools are needed to guide interdisciplinary management processes [7].

The literature recommends several instruments for the early diagnosis of CP, including the Hammersmith Infant Neurological Examination (HINE) [8], the WHOQOL-BREF quality of life questionnaire, and the Comprehensive Neonatal Speech-Language Assessment Protocol (EFIN).

However, in Colombia—and specifically in Medellín—these tools are not yet integrated into evaluation and follow-up protocols. Likewise, no studies have documented interdisciplinary follow-up of preterm children and their families, or explored its impact on functionality, feeding, participation, and long-term quality of life.

Based on the above, this study sought to analyze the relationship between neuromotor function, primary orofacial functions (sucking-swallowing-breathing), and early communication in preterm newborns with a corrected age of 2 to 3 months. In addition, the perceived quality of life of the primary caregiver in a hospital center in Medellín was assessed.

## 2. Materials and Methods

### 2.1. Study Characteristics and Participants

A quantitative, cross-sectional, analytical, observational study was conducted at a hospital center in Medellín, Colombia. The aim was to analyze, in a group of premature infants with a corrected age between 2 and 3 months, the relationship between neurological risk and primary stomatognathic functions (sucking, swallowing, and breathing), early communication, and the perceived quality of life of their caregivers. Sociodemographic data and relevant clinical history were also collected. We used a single-center design to ensure uniform clinical protocols (consistent implementation of the KFP), the same trained assessment team (HINE and EFIN), and standardized procedures, thereby reducing between-site heterogeneity and enhancing internal validity in this initial phase.

The study population consisted of premature infants enrolled in the Kangaroo Family Program (KFP), a care pathway for premature infants after discharge. A non-probabilistic convenience sampling method was applied, including all premature infants who attended the service and met the inclusion criteria during the recruitment phase. Enrollment in the KFP, the institutional standard of care, was required to ensure homogeneous clinical follow-up, access to standardized medical records, consistent breastfeeding support, and uniform caregiver education.

Inclusion criteria included infants born before 37 weeks of gestation, with a corrected age between 2 and 3 months at the time of assessment, who were hemodynamically stable. In addition, their legal guardians had to provide signed informed consent, and the infants had to be enrolled in the Kangaroo Family Program at the Clínica Universitaria Bolivariana during the second semester of 2023 and the first semester of 2024 in Medellín. Exclusion criteria included hemodynamically unstable infants whose condition posed a life risk, those with congenital malformations, or cases in which participation was not accepted or consent was not provided.

The surveys and measurement tools were applied between September 2023 and May 2024. The study was approved by the Ethics Committee of Fundación Universitaria María Cano (Record No. 2, 23 June 2023). Evaluations were conducted in accordance with the guidelines and approval of the Scientific Committee of the Pontificia Bolivariana Clinic (Record No. 15, 17 July 2023), and followed Resolution 008430 of the Colombian Ministry of Health and Social Protection.

### 2.2. Instruments

The surveys and measurement tools were administered in person at the facilities of the Clínica Universitaria Bolivariana. Sociodemographic and clinical data were collected by physiotherapists trained in standardized chart abstraction and caregiver interviewing. A structured survey was used to obtain sociodemographic and clinical data, including sex, socioeconomic status, affiliation with the General System of Social Security (SGSS), and household conditions. A semi-structured survey was also applied to gather clinical data such as chronological and corrected age, birth weight, height, head circumference, length of hospitalization, oxygen therapy use and duration, and the presence of clinical conditions such as Patent Ductus Arteriosus (PDA), bronchopulmonary dysplasia, necrotizing enterocolitis, Retinopathy of Prematurity (ROP), as well as major hospital complications including high-grade Intraventricular Hemorrhage (IVH), Periventricular Leukomalacia (PVL), and hypoxic–ischemic encephalopathy.

Neurological function was assessed using the Hammersmith Infant Neurological Examination (HINE). The HINE consists of 26 items that evaluate cranial nerve function, posture, quality and quantity of movement, tone, and reflexes/reactions. Each item is scored on a scale from 0 to 3, with a total score ranging from 0 to 78. The tool has been standardized for very preterm infants aged 6 to 15 months corrected age, and cut-off scores have been validated in populations of preterm infants between 3 and 12 months post-term. In typically developing infants, optimal scores are defined as those achieved in 90% of cases. Motor milestones are also recorded during follow-up visits using the second part of the HINE [9]. According to Novak et al. [10], scores below 57 at 3 months are 96% predictive of CP; scores below 73 at 6, 9, or 12 months indicate high risk, and scores below 40 almost always indicate CP. The HINE was administered by physiotherapists trained in pediatric neurodevelopment and certified in the application of the scale by the Mac Keith Press.

The WHOQOL-BREF questionnaire, developed by the World Health Organization, includes 26 items assessing quality of life and health satisfaction across four domains: physical health, psychological health, social relationships, and environment [11]. Raw scores were calculated for each domain and interpreted as follows: 0–20 (poor quality of life), 21–40 (moderate), 41–60 (good), and 61–80 (very good). According to Cardona et al. [12], the instrument has excellent psychometric properties, with 100% internal consistency and discriminant validity across all dimensions. The strongest correlations (*p* ≥ 0.4) were observed in physical health, psychological health, and mental health [12]. The WHOQOL-BREF was administered to caregivers in a quiet room; when literacy barriers or comprehension difficulties arose, a trained psychologist provided reading support without influencing responses.

The Comprehensive Neonatal Speech-Language Assessment Protocol (EFIN) was used to evaluate the primary stomatognathic functions of neonates in order to identify the minimum conditions required for successful breastfeeding and early mother–infant communication. The protocol includes five evaluation dimensions, each with specific items assessing the ideal parameters of sucking–swallowing–breathing coordination, early communication, and breastfeeding [13]. Each dimension is scored on scales ranging from 0 to 1, 0 to 2, or 0 to 3, depending on the number of items. The lowest scores indicate altered conditions, while higher scores indicate expected performance. The maximum score is 70; scores between 65 and 70 reflect adequate development of the sucking–swallowing–breathing triad, whereas scores below 68 indicate alterations in this triad. The EFIN was conducted by a speech-language pathologist experienced in neonatology. All assessments were performed in a single visit per mother–infant dyad, ensuring privacy, appropriate lighting, and comfort.

### 2.3. Statistical Analysis 

Data analysis was performed using RStudio (version 4.4.2) as the interface for the R Project software (version 4.4.2), along with Python (version 3.12.5) for data modeling. Exploratory data analysis, visualizations, and data quality reports were carried out in R. For data modeling, several Python modules were employed. Pearson’s correlation coefficient was used to identify linear relationships between covariates. Algorithms such as Support Vector Machines (SVM), Linear Regression, XGBoost, and Random Forest were applied for modeling. Model performance was evaluated using Mean Squared Error (MSE) and the coefficient of determination (R^2^) to assess accuracy and explanatory power.

## 3. Results

A total of 58 neonates were referred from the institution’s Kangaroo Family Program. However, after applying the study’s inclusion criteria, 15 were excluded because they were not premature and had no relevant risk factors, resulting in a final sample of 43 participants.

Regarding sociodemographic characteristics, 58.14% of the participants were male. More than half (51.16%) belonged to socioeconomic stratum 2 (low), indicating that most families of the neonates face limitations in accessing basic public services. In addition, 53.49% were covered under the contributory health insurance scheme through their mothers. Most families (81.49%) reported access to all basic services at home, and the majority of participants (67.44%) resided in Medellín (Table 1).

For the clinical data, a total of 36 preterm infants and 7 full-term infants were included based on gestational age at birth. The latter were included due to the presence of significant neurological risk factors. The gestational age of the newborns ranged from 26 to 40 weeks (Table 2).

Among the preterm infants, the corrected age at the time of evaluation ranged from 3 to 108.5 days (3 months and 2 weeks). Regarding head circumference, the average was 31.6 cm for males and 30.7 cm for females (Table 3).

In terms of anthropometric measurements, the infants had an average birth weight below 2246 g, indicating low birth weight. The average length at birth was 43.6 cm for females and 44.6 cm for males (Table 4). Regarding APGAR scores at 5 min, 90.69% scored between 7 and 10, indicating no signs of perinatal hypoxia, while only one infant had a score below 4. However, 46.51% presented some complication during delivery, and 60% were born via vaginal delivery.

With regard to complications associated with prematurity, 33 infants required postnatal hospitalization, with lengths of stay ranging from 1 to 60 days. A total of 44.18% of the neonates experienced respiratory difficulties. On average, male infants required 18.6 days of oxygen therapy compared with 8.7 days for females, with a maximum of 51 days. However, only 34.88% of the sample required mechanical ventilation (Table 4).

In addition, 10 infants required surfactant during the first days after birth. Of these, it was administered to 9 preterm neonates (<37 weeks) and to 1 full-term neonate. In the latter case, its use could be associated with an APGAR score below 3, as no other detailed report was available in the clinical history to justify its administration. Finally, 7 neonates developed sepsis, and 1 was diagnosed with patent ductus arteriosus.

Neonatal morbidities—including PDA, bronchopulmonary dysplasia, necrotizing enterocolitis, ROP, IVH, PVL, and hypoxic–ischemic encephalopathy—were verified through medical record review using a standardized form. None of the infants included in the study had a documented history of these conditions.

Neurological risk assessment with the HINE showed that only 9.30% of premature neonates scored below 57 points, primarily males, indicating a possible risk of cerebral palsy. The lowest-scoring domain was movement quality (Table 5).

In the EFIN assessment, 97.67% of participants scored above 68 points, indicating no alterations in the development of the sucking–swallowing–breathing triad. However, the breastfeeding component had the lowest scores among all evaluated dimensions (Table 6). 

With respect to the WHOQOL-BREF, caregivers reported the lowest perceived quality of life scores in the domain of social relationships, followed by psychological health. These impacts were more pronounced in families of male neonates (Table 7).

Regarding the overall results of the three scales, seven neonates scored below 57 points on the HINE. Among them, five were preterm, two had received surfactant, one experienced mild perinatal hypoxia, and two were delivered by emergency cesarean section.

For the EFIN, only one neonate scored below 68 points, with prematurity as the only relevant background factor.

Finally, in terms of overall quality of life results according to the WHOQOL-BREF, 18 family members reported the lowest scores (3/5) in the total sample, with wide variability in the associated risk factors, as shown in Table 8.

Finally, Pearson correlation analysis revealed no strong linear relationships between variables. None of the associations reached statistical significance (*p* ≤ 0.05), as shown in Figure 1.

## 4. Discussion

The results highlight the importance of implementing institutional protocols for the comprehensive evaluation of premature newborns and their families. In this regard, [14] emphasize that the intervention of an interdisciplinary team is essential to provide adequate follow-up for these children, facilitating the early detection of problems that, although not always evident in the first years of life, can affect their development and health at later stages. This allows a more accurate prognosis to guide care and intervention. Complementing this, Toro-Huerta et al. [15] explored environmental factors in the Colombian child population, relating them to the sociodemographic data for the Antioquia region described in this study. Both studies highlight the importance of understanding socioeconomic context and access to services as key determinants of the well-being of newborns and their families [15]. In addition, the adequate coverage of basic services at home (81.49%) observed in this study is a positive factor for child health, as it supports optimal development and growth.

Regarding the clinical and anthropometric data obtained from premature neonates, gestational age at birth ranged from 26 to 37 weeks, reflecting a population with high variability in intrauterine development, consistent with the prematurity profile. A significant finding was that 90.69% of neonates had APGAR scores between 7 and 10 at 5 min, indicating adequate initial adaptation in most cases without signs of perinatal hypoxia. However, the fact that 46.51% experienced some complication during delivery, and that 60% were born vaginally, suggests that perinatal factors must be carefully evaluated to prevent future complications.

According to postpartum hospitalization data, a considerable proportion of neonates required prolonged hospital stays, with respiratory complications emerging as a common factor. This emphasizes the need for interventions such as oxygen therapy. Even so, only 34.88% required mechanical ventilation, which could reflect care practices that help minimize its use. This is how in the research of Mira et al. [16] emphasizes that mothers may experience significant depressive symptoms and high stress levels related to hospitalization, which negatively affect dyadic interactions such as physical contact and communication.

The use of surfactant, the incidence of neonatal sepsis, and other complications such as patent ductus arteriosus, although less frequent, underscore the complexity of managing these newborns. Notably, no cases of retinopathy of prematurity, intraventricular hemorrhage, or periventricular leukomalacia were observed, which could be related to advances in neonatal care that reduce the risk of these complications. However, the frequency of interventions and complications reinforces the need to continue improving the monitoring and management of premature neonates to optimize long-term outcomes.

Results from the HINE showed that only 9.30% of premature neonates scored below 57 points, indicating that most achieved acceptable results on this evaluation. This may reflect a favorable trend in early neuromotor development. A notable finding was the predominance of males in the group with lower scores, indicating a greater susceptibility among premature male neonates to neuromotor difficulties or to being at risk of diagnoses such as cerebral palsy. This observation is consistent with a previous study [17] reporting that people with cerebral palsy exhibit impairments in motor function, general stability, and coordination, as well as associated digestive disturbances manifesting as gastrointestinal disease, poor nutritional status, and gastroesophageal reflux, among others.

Most premature neonates evaluated did not present alterations in the development of the sucking–swallowing–breathing triad, which is consistent with previous studies such as [18]. In fact, 97.67% of neonates in this study achieved scores above 68 on the EFIN scale, indicating a favorable progression in most cases. However, the breastfeeding component obtained the lowest scores compared with other aspects of the triad. Breastfeeding, which requires complex coordination of sucking, swallowing, and breathing, can be particularly challenging for premature neonates due to their lower neuromuscular development and reduced energy reserves.

Most neonates evaluated showed no signs of dysphagia, since orofacial functions activate the stomatognathic system to initiate, sustain, and ensure effective and safe feeding. Proper positioning and attachment of the lips, tongue, and palate, together with synchronized movements, promote safe swallowing. This finding aligns with Vijay et al. [19], who reported that adequate oral placement of stomatognathic structures in preterm and term neonates supports nutritional recovery, weight gain, and maturation of oromotor skills. Although breastfeeding is considered safe from birth through six months, intrinsic and extrinsic factors influence early experiences of swallowing.

Communication, a higher-order mental function, is innate and develops from the womb, where the baby is already able to recognize the parents’ voices. At birth, if there are no complications, the first connection is established during breastfeeding with the mother, and this bond continues through the dyad. It is important to guide parents on how to use this space, informing them about communicative acts such as crying, touch, eye contact, and body movements to strengthen communication and its development. This is consistent with [20], who emphasized that environmental stimuli influence child development and that, through both positive and negative experiences and beliefs, cognitive development is stimulated.

Results from the WHOQOL-BREF revealed lower perceptions of quality of life in aspects related to social relationships and psychological health, underscoring the importance of these factors in the well-being of primary caregivers of neonates. This finding is consistent with [21], which suggests that social relationships and psychological well-being play a fundamental role in the perception of quality of life, particularly in contexts of stress or vulnerability. An additional finding was the greater impact observed in families with male neonates. This could be related to gender differences in child health, where families may perceive greater challenges or concerns in raising a male child due to social expectations, gender roles, or biological differences.

In general, these results emphasize the importance of addressing both social and psychological aspects of health in support programs for families of neonates, promoting strategies to strengthen social relationships and emotional well-being. This is crucial not only to improve individual perceptions of quality of life but also to provide a supportive environment that helps families cope with the challenges of raising a premature newborn. The interaction of social and psychological factors highlights the need for comprehensive interventions that address not only the physical health of the neonate but also the mental and social health of caregivers. These findings suggest that future research should further explore the interaction of these factors and how gender differences, along with social expectations, can influence the overall experience of quality of life in childrearing.

Some methodological limitations should be noted, such as including only neonates enrolled in the KFP. While this design choice enhances internal validity by reducing variability in care, it may limit external generalizability to settings without KFP. Operational constraints (recruitment timelines, ethics approvals, and available resources) also shaped the feasibility and follow-up of participants. A multicenter expansion is planned to examine the generalizability of these findings.

## 5. Conclusions

The findings indicate that sociodemographic, clinical, and emotional factors influence the development of preterm newborns and the quality of life of their caregivers. This underscores the need for a multidisciplinary approach that integrates both clinical care and psychosocial support. Future research should further explore the role of gender and social conditions in parenting experiences and family well-being.

## Figures and Tables

**Figure 1 children-12-01483-f001:**
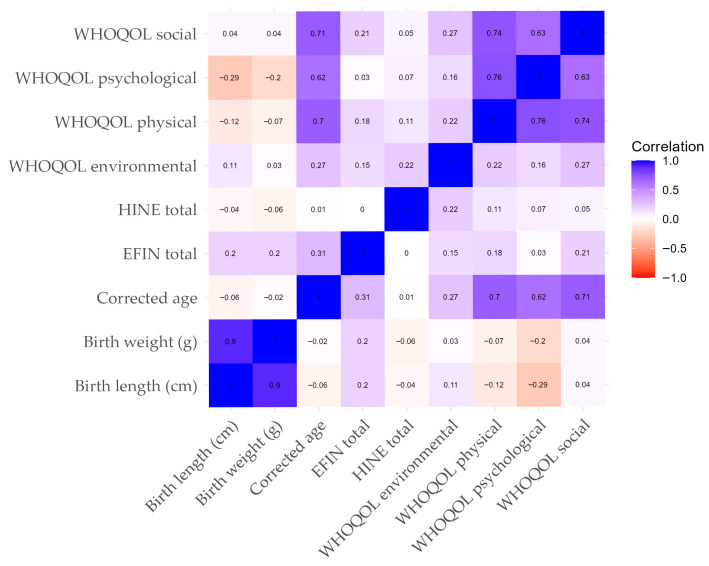
Correlation analysis between sociodemographic variables and the HINE, EFIN, and WHOQOL scales.

**Table 1 children-12-01483-t001:** Sociodemographic characteristics of premature neonates and their families included in the study.

Sociodemographic Variable	Categories	Frequency	Percentage (%)
Sex	Male	25	58.14
	Female	18	41.86
Socioeconomic Stratum	2 (low)	22	51.16
	1 (low-low)	15	34.88
	3 (lower-middle)	4	9.30
	0 (unstratified)	2	4.65
SGSSS Affiliation	Contributory	23	53.49
	Subsidized	19	44.19
	Uninsured	1	2.33
Household Conditions	All basic services available	35	81.40
	Missing one or more basic services	7	16.28
	No basic services	1	2.33
City of Residence	Medellín, Antioquia	29	67.44
	Bello, Antioquia	3	6.97
	La Estrella, Antioquia	3	6.97
	Zona rural Chocó	2	4.65
	Andes, Antioquia	1	2.32
	Amaga, Antioquia	1	2.32
	Caldas, Antioquia	1	2.32
	Itagüí, Antioquia	1	2.32
	Maceo, Antioquia	1	2.32
	Santo Domingo, Antioquia	1	2.32

**Table 2 children-12-01483-t002:** Gestational age of the total number of neonates included in the study.

Gestational Age	Frequency	Percentage (%)
26–28 weeks	1	2.33
29–31 weeks	6	13.95
32–34 weeks	9	20.93
35–37 weeks	20	46.51
38–40 weeks	7	16.28

**Table 3 children-12-01483-t003:** Anthropometric measurements according to gestational age of the neonates included in the total sample.

GA	MW	SD W	Q1 W	Q2 W	Q3 W	ML	SD L	Q1 L	Q2 L	Q3 L	M HC	SD HC	Q1 HC	Q2 HC	Q3 HC
26	1108		1108	1108	1108	36		36	36	36	30		30	30	30
29	1141.67	206.54	1022.5	1030	1205	36	1.73	35.5	37	37	26.67	3.21	25.5	28	28.5
30	1450		1450	1450	1450	40		40	40	40	27		27	27	27
31	1740	254.56	1650	1740	1830	41	4.24	39.5	41	42.5	31.25	1.77	30.63	31.25	31.88
32	959		959	959	959	37		37	37	37	27		27	27	27
34	1973.15	833.24	2002.75	2110	2491	43.88	2.1	42	44	45.25	31.88	1.25	31	32	33
35	2184.5	133.64	2137.25	2184.5	2231.75	44.5	3.54	43.25	44.5	45.75	31	1.41	30.5	31	31.5
36	2532.56	238.06	2326	2515	2680	46.78	1.79	45	46	48	32.22	0.79	32	32.5	33
37	2299.78	148.43	2180	2216	2460	45.67	1.87	44	47	47	31.61	1.5	30	31.5	33
38	2392.5	141.04	2337.5	2405	2460	45.75	1.26	45.5	46	46.25	32	0.82	31.75	32	32.25
39	2192.5	272.24	2096.25	2192.5	2288.75	47	0	47	47	47	32	1.41	31.5	32	32.5
40	3410		3410	3410	3410	50		50	50	50	36		36	36	36

GA = Gestational age (weeks); MW = Mean weight (g); SD = Standard Deviation; W = Weight; ML = Mean Length (cm), L = Length; MHC = Mean Head Circumference (cm); HC = Head Circumference (cm).

**Table 4 children-12-01483-t004:** Analysis of Quantitative Clinical Variables of Premature Neonates Included in the Study.

Variable	25%		50%		75%		Maximum		Mean		Minimum		SD	
Sex	M	F	M	F	M	F	M	F	M	F	M	F	M	F
Hospitalization days	5	3.25	8	6.5	12	28.75	60	60	13.4	19.4	2	1	15.37	23.4
Oxygen therapy days	3.25	1	10	2	30.7	8	51	50	18.6	8.7	1	0	19.02	15.8
Corrected age (days)	35	51.5	51	66	90	79.5	108.5	104	102.8	64.7	3	28	211.3	21.1
Head circumference (cm)	31	30	32	31.5	33	32.5	36	33	31.6	30.7	27	23	1.9	2.54
Birth weight (g)	2060	2035.2	2290	2198	2515	2435.7	3410	2711	2246.9	2083.1	1030	959	507.2	523.9
Birth length (cm)	44	42	46	44	47	47	50	50	44.6	43.6	37	34	3.4	4.3

Note: M = Male; F = Female; SD = Standard Deviation.

**Table 5 children-12-01483-t005:** Results of the neurological examination using the HINE scale in premature neonates assessed by the physiotherapy team.

Variable	25%		50%		75%		Maximum		Mean		Minimum		SD	
	M	F	M	F	M	F	M	F	M	F	M	F	M	F
HINE—Movements	6	6	6	6	6	6	6	6	5.56	5.67	1	3	1.16	0.84
HINE—Cranial Nerves	14	13	15	15	15	15	15	15	14.32	14.11	10	12	1.46	1.32
HINE—Posture	11	13	15	14.5	16	15.75	18	18	13.68	14.28	7	10	3.11	2.16
HINE—Reflex	7	9	9	10	11	11	15	15	9.24	9.83	5	5	2.70	2.18
HINE—Muscle Tone	19	19.5	21	21.5	23	22.75	24	24	20.68	21.17	10	16	3.16	2.23
HINE—Total	58	63.25	67	65	70	67.75	78	72	63.48	65.06	33	55	9.22	4.40

Note: M = Male; F = Female; SD = Standard Deviation.

**Table 6 children-12-01483-t006:** Results of the Sucking–Breathing–Swallowing triad evaluation using the EFIN scale in premature neonates assessed by the speech-language pathology team.

Variable	25%		50%		75%		Maximum		Mean		Minimum		SD	
	M	F	M	F	M	F	M	M	M	F	M	F	M	F
EFIN—Communication	13	11.25	15	13.5	15	15	15	15	13.48	13.22	10	10	1.90	1.93
EFIN—Functional evaluation	42	43.25	44	45	45	45	45	45	43.16	43.44	37	34	2.49	2.85
EFIN—Breastfeeding	10	10.25	11	11	11	11	11	11	9.64	9.61	0	3	2.80	2.79
EFIN—Orofacial Functions	21	21.25	22	22	22	22	22	22	21.16	21.22	19	16	1.21	1.63
EFIN—Total score	84	87.25	90	89.5	92	92	93	93	87.44	87.50	76	65	5.68	7.29

Note: M = Male; F = Female; SD = Standard Deviation.

**Table 7 children-12-01483-t007:** Results of the Quality of Life Assessment Using the WHOQOL-BREF Scale in Premature Neonates Assessed by the Psychology Team.

Variable	25%		50%		75%		Maximum		Mean		Minimum		SD	
	M	F	M	F	M	F	M	M	M	F	M	F	M	F
WHOQOL—Environmental Health	14	13.25	17.5	15.5	19	17	20	20	16.42	15.61	10.5	11	3.00	2.79
WHOQOL—Psychological Health	14.7	15.48	17.3	17	18.7	18	20	20	16.46	16.81	8	12	3.26	2.18
WHOQOL—Social Relationships	14.7	14.7	16	17.3	18.7	18.35	20	20	16.06	16.14	4	6.7	3.70	3.44
WHOQOL—Physical Health	15.4	15.03	16.6	16	18.3	17.55	20	18.9	16.19	16.09	10.3	13.1	2.73	1.82

Note: M = Male; F = Female; SD = Standard Deviation.

**Table 8 children-12-01483-t008:** Comparison between gestational age, postpartum complications, and overall results of the HINE, EFIN, and WHOQOL-BREF scales.

GA	Surfactant	APGAR 5 min	DH	DO	TD	HINE	EFIN	WHOQOL-BREF QL
36	No	7–10	4	0	Vaginal	72	82	3/5
37	No	7–10	5	0	Vaginal	65	93	4/5
34	Yes	7–10	0	0	Cesarean Emergencies	53	92	5/5
38	No	7–10	2	0	Vaginal	67	92	3/5
34	No	7–10	4	2	Vaginal	64	93	5/5
35	Yes	7–10	60	0	Vaginal	59	65	3/5
35	No	7–10	3	0	Vaginal	62	77	3/5
37	No	7–10	5	0	Cesarean Scheduled	33	84	4/5
29	Yes	7–10	31	31	Cesarean Emergencies	67	85	5/5
34	No	7–10	16	9	Cesarean Emergencies	62	89	5/5
37	Yes	7–10	7	0	Cesarean Scheduled	67	92	5/5
37	No	7–10	0	0	Cesarean Emergencies	65	90	5/5
34	No	7–10	10	10	Vaginal	71	92	3/5
30	No	7–10	0	51	Cesarean Emergencies	68	84	3/5
37	No	7–10	3	0	Cesarean Emergencies	68	83	3/5
36	No	7–10	8	0	Vaginal	70	93	5/5
37	No	7–10	5	0	Vaginal	55	80	3/5
36	No	7–10	0	0	Vaginal	67	85	4/5
36	No	7–10	0	0	Vaginal	72	93	5/5
34	No	7–10	6	1	Vaginal	56	76	3/5
37	No	7–10	6	2	Vaginal	71	87	4/5
38	No	7–10	0	0	Cesarean Emergencies	56	82	5/5
34	No	7–10	19	1	Vaginal	67	93	3/5
32	No	7–10	33	7	Cesarean Emergencies	65	93	5/5
39	No	7–10	12	0	Vaginal	69	91	5/5
40	Yes	4–3	50	47	Vaginal	78	89	5/5
36	No	7–10	1	0	Vaginal	63	89	3/5
36	Yes	7–10	0	0	Cesarean Emergencies	70	89	5/5
38	No	7–10	10	0	Vaginal	71	93	3/5
36	No	7–10	0	0	Vaginal	60	85	3/5
39	No	7–10	0	0	Vaginal	61	93	5/5
29	No	6–5	31	10	Cesarean Emergencies	69	93	3/5
36	No	7–10	0	0	Vaginal	59	93	5/5
31	Yes	7–10	12	4	Vaginal	72	78	5/5
38	No	6–5	3	0	Vaginal	55	90	3/5
37	No	7–10	3	0	Cesarean Scheduled	70	92	5/5
34	No	7–10	60	3	Cesarean Scheduled	58	87	5/5
31	Yes	6–5	38	31	Cesarean Emergencies	63	91	5/5
26	Yes	7–10	60	74	Vaginal	64	90	3/5
29	No	7–10	60	8	Vaginal	64	74	3/5
37	No	7–10	6	0	Cesarean Scheduled	62	91	5/5
36	No	7–10	0	0	Cesarean Emergencies	72	88	3/5
34	Yes	7–10	8	1	Vaginal	56	90	5/5

GA: Gestational age (weeks); DH: Days of hospitalization; DO: Days of oxygen therapy; TD: Type of delivery; QL: Quality of life.

## Data Availability

Data supporting the findings of this study are available from the corresponding author upon reasonable request due to legal and ethical considerations.

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
