# Peer review of "Interdisciplinary Assessment of Premature Newborns and Their Families in a Hospital Setting in Medellín, Colombia"

_children, 2025, doi:10.3390/children12111483_

Round 1

Reviewer 1 Report

Comments and Suggestions for Authors

1- INTRODUCTION
- Sentences that are too long and complex
- LINE 44: Since this is the first time the term appears, they should include "cerebral palsy (CP)"
- LINE 47: Hypoxia is not the only factor that can cause cerebral palsy in premature children (Rees P, Callan C, Chadda KR, Vaal M, Diviney J, Sabti S, Harnden F, Gardiner J, Battersby C, Gale C, Sutcliffe A. Preterm Brain Injury and Neurodevelopmental Outcomes: A Meta-analysis. Pediatrics. 2022 Dec 1;150(6):e2022057442)

2 MATERIALS AND METHOD
They should describe how the questionnaires were administered: who was the professional in charge (were they psychologists, pediatricians, etc.), where the questionnaires were administered...

3 RESULTS
- They should describe how the sample size was calculated. For this type of article, 43 children seems like a small sample size.
- Explanations regarding gestational age at birth should be provided: the number of children < 28 weeks, 28-32 weeks, 32-34 weeks, and 34-37 weeks. It is not possible to jointly study the psychomotor development or weight-height of children of such disparate gestational ages.
The reduced number of children who required surfactant (10) suggests that very few children with gestational ages < 32 weeks were included, who are precisely those most at risk of future neurological problems.
- Given that the group is made up of children of different gestational ages, the values given for weight, height, and head circumference are of no help. In any case, the mean, standard deviations, and  quartiles of PERCENTIL according to gestational age should be reflected-

- Table 1: Define 0-1-2 and 3 values for the Socioeconomic stratum.
- Table 3, 4 and 5: Characteristics of children with the worst outcomes: gestational age, problems during admission.

4- DISCUSSION
- References are made to results not explained in the previous section. For example, the socioeconomic status of each family is not analyzed, and yet paragraphs to this effect are included in the discussion (lines 215-234). Children with periventricular hemorrhage retinopathy are also not mentioned in the results (line 258 onward).
- There is no explanation as to why the outcome is worse in boys.

5 CONCLUSIONS:
Only the last paragraph is necessary.

Comments on the Quality of English Language

The mail problem with Quality of English Language is that sentences are too long and complex

Reviewer 2 Report

Comments and Suggestions for Authors

Thank you for sending your article to our journal.  
Firstly, I would like to congratulate the team of researchers for working on an important subject and conducting an intricate statistical analysis.  
The following are comments, questions, or suggestions that may need to be addressed to make this an even better manuscript for publication.

  1. Suggest to use a grammar checking software to proof read the article. There are many errors in the sentence framing.
  2. The introduction needs more trimming, and relevant articles to be quoted.  
  3. In the methods section, there was a mention that data regarding NICU morbidities was collected (PVL, IVH etc). None of these are represented in the results.
  4. All neonates below 37 weeks were included in the study and the lowest gestational age was 26 weeks. Assessment of a 26 week neonate and a 36 weeks neonate significantly differs with respect to the neurological outcomes. It would have been ideal if there were some subgroups with respect to the gestational age,
  5. There was a mention that there were no IVH, BPD, ROP in the study - this depends on the number of extreme preterm neonates in the group. It is surprising to see that there were no morbidities in the study population
  6. Baseline characteristics should include the gestational age along with other relevant variables/morbidities
  7. Discussion needs to be significantly trimmed. 
  8. Conclusion should be very precise and should be limited to the a maximum of 2-3 sentences highlighting the findings in the study. These statements should be aligned with the primary and secondary outcomes
Comments on the Quality of English Language

The quality of the English should be improved significantly and I would suggest using a grammar checking software to correct the errors.

Many sentences are too lengthy which do not accurately convey the intended meaning. 

Reviewer 3 Report

Comments and Suggestions for Authors

listed in the document 

Reviewer 4 Report

Comments and Suggestions for Authors

The authors need to be complimented for a good study on a topic of immense relevance to neonatology.Certain observations on the manuscript are as follows:-

(a) There are few grammatical errors that need to be corrected

(b) The study design could have incorporated a multicentric model in order to introuduce heterogeneity of subjects 

(c) The methodology otherwise is robust

(d) Though the relevance of using scoring systems for optimal neonatal care is proved and is advocated by major neonatological bodies internationally, still, the study has been performed in a geographical locale where these scores have not been routinely applied currently.Hence the study is relevant .

Comments on the Quality of English Language

There are few grammaticalerrors which need to be corrected.

Round 2

Reviewer 1 Report

Comments and Suggestions for Authors

Once the corrections have been made, the article will be ready for publication.

Reviewer 2 Report

Comments and Suggestions for Authors

The originality and novelty are lacking in this study and the quality of the presentation is not acceptable. 

Comments on the Quality of English Language

The English language and grammar can be improved